# Agonist-selective recruitment of engineered protein probes and of GRK2 by opioid receptors in living cells

Miriam Stoeber[1,2]*, Damien Jullié[1,3], Joy Li[1,3], Soumen Chakraborty[4,5], Susruta Majumdar[4,5], Nevin A Lambert[6], Aashish Manglik[7,8], Mark von Zastrow[1,3]*

[1]Department of Psychiatry, University of California, San Francisco, San Francisco, United States; [2]Department of Cell Physiology and Metabolism, University of Geneva, Geneva, Switzerland; [3]Department of Cellular and Molecular Pharmacology, University of California, San Francisco, San Francisco, United States; [4]Center for Clinical Pharmacology, Washington University School of Medicine, St. Louis, United States; [5]St Louis College of Pharmacy, St. Louis, United States; [6]Department of Pharmacology and Toxicology, Medical College of Georgia, Augusta University, Augusta, United States; [7]Department of Pharmaceutical Chemistry, University of California, San Francisco, San Francisco, United States; [8]Department of Anesthesia, University of California, San Francisco, San Francisco, United States

*For correspondence:
miriam.stoeber@unige.ch (MS);
mark@vzlab.org (MZ)

**Competing interests:** The authors declare that no competing interests exist.

**Abstract** G protein-coupled receptors (GPCRs) signal through allostery, and it is increasingly clear that chemically distinct agonists can produce different receptor-based effects. It has been proposed that agonists selectively promote receptors to recruit one cellular interacting partner over another, introducing allosteric 'bias' into the signaling system. However, the underlying hypothesis - that different agonists drive GPCRs to engage different cytoplasmic proteins in living cells - remains untested due to the complexity of readouts through which receptor-proximal interactions are typically inferred. We describe a cell-based assay to overcome this challenge, based on GPCR-interacting biosensors that are disconnected from endogenous transduction mechanisms. Focusing on opioid receptors, we directly demonstrate differences between biosensor recruitment produced by chemically distinct opioid ligands in living cells. We then show that selective recruitment applies to GRK2, a biologically relevant GPCR regulator, through discrete interactions of GRK2 with receptors or with G protein beta-gamma subunits which are differentially promoted by agonists.

## Introduction

G protein-coupled receptors (GPCRs) comprise nature's largest family of signaling receptors and an important class of therapeutic drug targets. GPCRs signal by allostery, and were considered for many years to operate as binary switches that bind to cognate transducer and regulator proteins in a single agonist-induced activated state. Over the past decade an expanded view has taken hold, supported by accumulating in vitro evidence that GPCRs are conformationally flexible (*Lohse and Hofmann, 2015*; *Mahoney and Sunahara, 2016*; *Nygaard et al., 2013*; *Weis and Kobilka, 2018*; *Wingler et al., 2019*) and a confluence of cell biological and in vivo evidence supporting the existence of functionally selective agonist effects (*Smith et al., 2018*; *Urban et al., 2007*; *Williams et al., 2013*). According to this still-evolving view, agonists have the potential to promote GPCRs to selectively recruit one transducer or regulator protein over another, introducing bias into

**eLife digest** About a third of all drugs work by targeting a group of proteins known as G-protein coupled receptors, or GPCRs for short. These receptors are found on the surface of cells and transmit messages across the cell's outer barrier. When a signaling molecule, like a hormone, is released in the body, it binds to a GPCR and changes the receptor's shape. The change in structure affects how the GPCR interacts and binds to other proteins on the inside of the cell, triggering a series of reactions that alter the cell's activity.

Scientists have previously seen that a GPCR can trigger different responses depending on which signaling molecule is binding on the surface of the cell. However, the mechanism for this is unknown. One hypothesis is that different signaling molecules change the GPCR's preference for binding to different proteins on the inside of the cell. The challenge has been to observe this happening without interfering with the process.

Stoeber et al. have now tested this idea by attaching fluorescent tags to proteins that bind to activated GPCRs directly and without binding other signaling proteins. This meant these proteins could be tracked under a microscope as they made their way to bind to the GPCRs. Stoeber et al. focused on one particular GPCR, known as the opioid receptor, and tested the binding of two different opioid signaling molecules, etorphine and Dynorphin A.

The experiments revealed that the different opioids did affect which of the engineered proteins would preferentially bind to the opioid receptor. This was followed by a similar experiment, where the engineered proteins were replaced with another protein called GRK2, which binds to the opioid receptor under normal conditions in the cell. This showed that GRK2 binds much more strongly to the opioid receptor when Dynorphin A is added compared to adding etorphine.

These findings show that GPCRs can not only communicate that a signaling molecule is binding but can respond differently to convey what molecule it is more specifically. This could be important in developing drugs, particularly to specifically trigger the desired response and reduce side effects. Stoeber et al. suggest that an important next step for research is to understand how the GPCRs preferentially bind to different proteins.

the signaling cascade at a receptor-proximal level that is either propagated downstream or eliminated during intermediate transduction steps (*Lau et al., 2011*; *Tsvetanova et al., 2017*).

Opioid receptors provide a representative example. Interest in selective agonist effects at these GPCRs dates back to the initial demonstration that opioid receptors can be activated by diverse peptide and non-peptide agonists (*Kosterlitz and Hughes, 1977*). Early experimental evidence for such selectivity among ligands emerged from the observation of an agonist-induced state of opioid receptors in neuroblastoma cells that discriminates between opioid peptides and opiate alkaloids (*Von Zastrow et al., 1993*). This was followed by the demonstration of agonist-selective control of opioid receptor endocytosis, leading to the identification of functional selectivity among agonists defined by differences in relative ability to drive receptor engagement of G protein relative to beta-arrestin-dependent cellular pathways (*Keith et al., 1998*; *Keith et al., 1996*; *Whistler et al., 1999*; *Whistler and von Zastrow, 1998*). This concept further evolved to the present view of biased receptor recruitment of G proteins relative to beta-arrestins, with receptor-proximal selectivity calculated by fitting quantitative measures of downstream pathway or protein response to operational models of receptor-effector coupling (*Schmid et al., 2017*).

Two key gaps persist in our present understanding. First, selective protein recruitment by GPCRs in intact cells remains largely calculated rather than directly observed. Accordingly, the understanding of receptor-proximal agonist bias is inherently limited by assumptions of the model used to calculate it (*Kenakin, 2018*; *Klein Herenbrink et al., 2016*). Indeed, and despite intense efforts motivated by interest in the therapeutic impact of biased agonist effects at opioid receptors (*Johnson et al., 2017*; *Schmid et al., 2017*; *Whistler et al., 1999*), significant challenges remain in reliably assessing selectivity of receptor-proximal protein recruitment based on downstream cell-based readouts (*Conibear and Kelly, 2019*). Second, challenges can arise even using cell-based assays that are direct. For example, multiple methods have been developed to detect GPCR interaction with beta-arrestins in intact cells (*Chen et al., 2012*; *Kim et al., 2017*). However, this binding

involves multiple biochemical steps and, in particular, it typically requires the receptor to undergo prior agonist-induced phosphorylation (*Eichel et al., 2018*; *Gurevich et al., 1995*; *Thomsen et al., 2016*). This has been clearly established for opioid receptors (*Whistler and von Zastrow, 1998*; *Zhang et al., 1998*), for which full interaction with beta-arrestin requires the receptor to be phosphorylated at multiple sites in the cytoplasmic tail through a defined sequence of agonist-dependent reactions which are catalyzed by distinct GPCR kinase (GRK) isoforms (*Chiu et al., 2017*; *Just et al., 2013*; *Lau et al., 2011*; *Miess et al., 2018*). Accordingly, beta-arrestin recruitment measured in such assays clearly reflects a process that is considerably more complex than allosteric selection by the receptor.

Here we describe an alternative approach to address these knowledge gaps. We delineate a cell-based method to simply assess selective protein recruitment by opioid receptors at the receptor-proximal level, taking advantage of two engineered protein folds established to bind agonist-activated GPCRs in intact cells without requiring or engaging other known cellular proteins (*Stoeber et al., 2018*; *Wan et al., 2018*). Using these engineered proteins comparatively as orthogonal receptor-interaction biosensors, we directly demonstrate selectivity in receptor-proximal protein recruitment elicited by various opioid agonists in living cells. We then show how the principle of receptor-proximal protein selection applies in a more complex manner to GRK2, a biologically relevant regulator.

## Results

### Comparative detection of direct protein recruitment by opioid receptors in living cells

Two agonist-activated opioid receptor complexes have been described in structural detail (*Figure 1A*), one bound to a nucleotide-free G protein heterotrimer and another to an active state-stabilizing nanobody (Nb) (*Huang et al., 2015*; *Koehl et al., 2018*). The receptor conformation resolved in each complex is similar but not identical, with Nb and G protein interactions involving distinct molecular contacts on cytoplasmic domains of the receptor. Nbs are inherently orthogonal to intracellular biochemistry but heterotrimeric G proteins engage multiple cellular proteins in addition to activated receptors. Thus we focused on mini-G (mG) proteins, engineered versions of the Ras-like domain of G protein alpha subunits which bind directly to activated GPCRs but are not known to engage other cellular proteins (*Nehmé et al., 2017*; *Wan et al., 2018*). We assessed binding to receptors in intact cells by redistribution of fluorescently labeled Nb or mG fusion proteins from the cytoplasm to the plasma membrane (*Figure 1B*).

For a mG probe we chose mGsi, derived from the Ras-like domain of Gs alpha but with nine residues at the distal C-terminus replaced by the corresponding residues from Gi alpha1. These C-terminal residues form a major determinant of G protein coupling specificity (*Conklin et al., 1993*) by folding into a helical structure (alpha-5 helix) that occupies the agonist-activated GPCR core (*Carpenter and Tate, 2017*; *Koehl et al., 2018*). Because Gs couples poorly to opioid receptors, we reasoned that a sensor derived from mGsi would primarily detect this interaction. For a Nb probe we selected Nb33, previously used to detect activated mu (MOR) and delta (DOR) opioid receptors in living cells (*Stoeber et al., 2018*). Nb33 shares receptor contact residues with Nb39, a close analog that has been resolved at high resolution in complex with activated MOR (*Huang et al., 2015*) and in a similar complex with activated kappa opioid receptor (KOR) (*Che et al., 2018*). Because cytoplasmic residues contacted by the Nb in these structures are largely distinct from those engaged by the G protein alpha-5 helix, we reasoned that the Nb-derived sensor has the potential to provide different allosteric information.

Fluorescent protein fusions of mGsi or Nb33 localized diffusely when expressed in the cytoplasm of HEK293 cells, and recruitment by receptors was monitored using total internal reflection fluorescence microscopy (TIR-FM) in cells co-expressing Flag-tagged KOR (*Figure 1C*). Importantly, HEK293 cells do not express endogenous opioid receptors or other opioid ligand binding sites, thereby providing a null genetic background on which to directly examine protein probe recruitment mediated specifically by the co-expressed receptor. We observed rapid and robust recruitment of mGsi by KOR upon application of the kappa-selective peptide agonist Dynorphin A (DynA, Dynorphin 1–17). Recruitment of mGsi was reversible because application of the high-affinity competitive

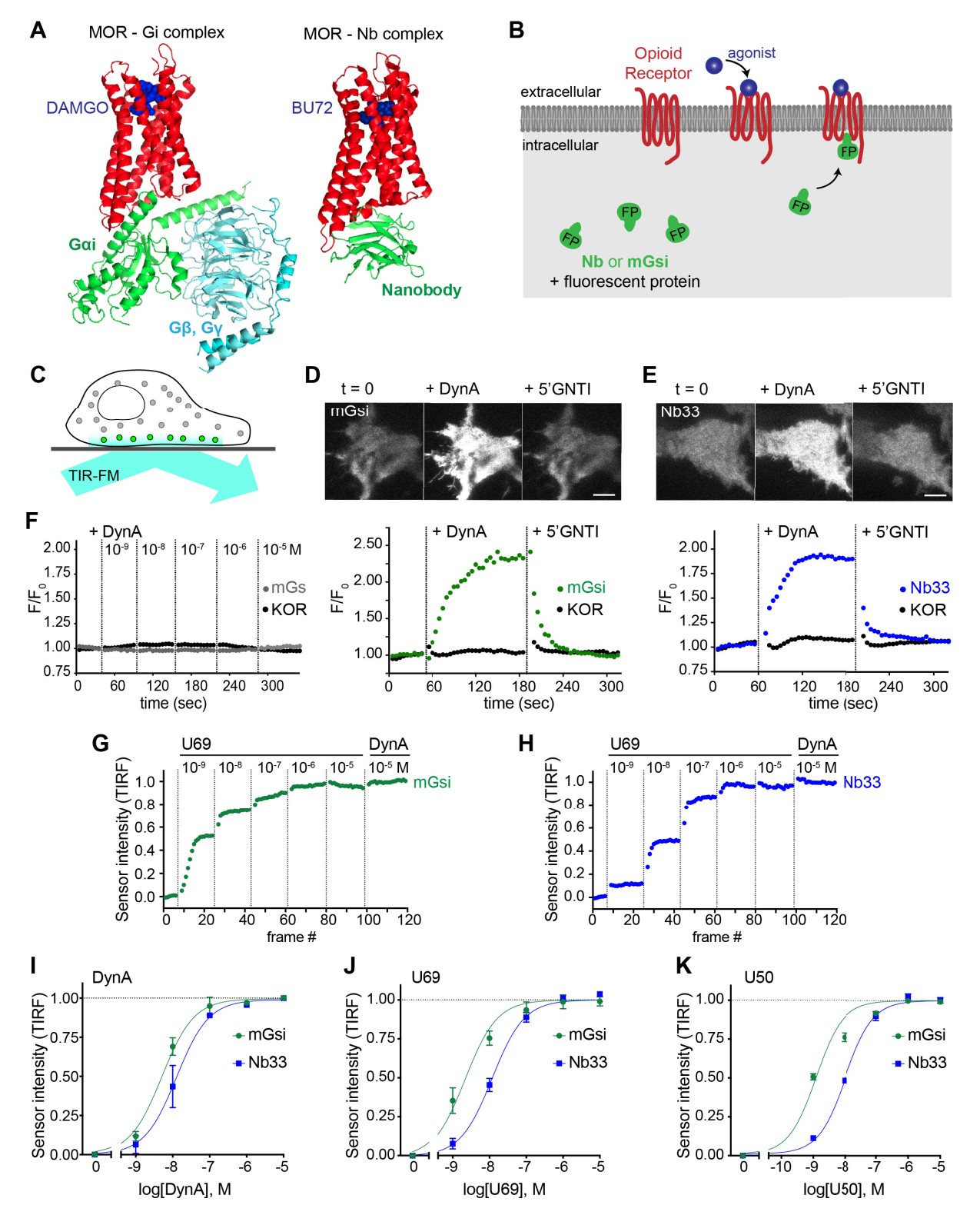

**Figure 1.** Comparative detection of direct probe recruitment by opioid receptors in living cells. (A) Crystal structures of the DAMGO-bound MOR (red) - Gi (green/blue) complex (PDB: 6DDF) and the BU27-bound MOR (red) – nanobody (green) complex (PDB: 5C1M). Ligands are shown in blue. (B) Schematic of nanobody (Nb)/miniGsi (mGsi) and OR localization in cells and expected probe re-localization upon agonist addition. (C) Scheme of a cell imaged by total internal reflection fluorescence microscopy (TIR-FM). The evanescent excitation field selectively illuminates fluorophores close to the

*Figure 1 continued on next page*

*Figure 1 continued*

plasma membrane. (D) TIR-FM images of a time series of a HEK293 cell, expressing Venus-mGsi and FLAG-KOR (not shown). Medium was exchanged to DynA (agonist, 100 nM) and to 5'GNTI (antagonist, 100 µM) by bath application. The scale bar represents 10 µm. Intensity of mGsi and KOR (labeled with anti-FLAG M1-AF647) during the TIR-FM time-lapse. 5 s between frames is shown. $F_0$, average fluorescence intensity before agonist. (E) Same as in (D) but with HEK293 cell expressing EGFP-Nb33 instead of mGsi. Intensity of Nb33 and KOR during TIR-FM time-lapse with 5 s between frames is shown. (F) Intensity of mGs and KOR (labeled with anti-FLAG M1-AF647) during the TIR-FM time-lapse, adding increasing concentrations of DynA (1 nM - 10 µM). 5 s between frames is shown. $F_0$, average fluorescence intensity before agonist. (G) mGsi intensity during TIR-FM time-lapse series of a HEK293 cell, co-expressing Venus-mGsi and KOR, adding increasing concentrations of U69 (1 nM - 10 µM) followed by reference compound DynA (10 µM). 5 s between frames is shown. Intensity is normalized between 0 (no agonist) and 1 (reference compound DynA). (H) Same as in (F) with HEK293 cell expressing EGFP-Nb33 instead of mGsi. (I–K) Concentration-dependent recruitment of mGsi and Nb33 probes to KOR, measured by TIR-FM upon different agonists. Normalization of intensity values is shown (range [0–1]). Regression curves with Hill slope of 1 are shown. (I) DynA concentration response (n = 3; average ± SEM). (J) U69 concentration response, normalized to DynA (n = 3; average ± SEM). (K) U50 concentration response, normalized to DynA (n = 4; average ± SEM).

The online version of this article includes the following source data for figure 1:

**Source data 1.** Concentration-dependent recruitment of mGsi and Nb33 probes to KOR in response to DynA, U69, and U50 (*Figure 1I-K*).

KOR antagonist 5'GNTI resulted in rapid redistribution of the biosensor back to the cytoplasm (*Figure 1D*). In contrast, mGs was not detectably recruited in response to KOR activation by DynA using the same assay (*Figure 1F*), verifying assay specificity and that mGsi recruitment is driven primarily by the Gi-derived distal C-terminus. Further, we verified that agonist-induced recruitment of mGsi occurred separately from a change in surface expression of KOR, which was monitored in parallel using anti-Flag antibody (*Figure 1D*). Nb33 was also rapidly recruited in response to KOR activation by DynA using the same experimental protocol, and this recruitment was also reversible upon antagonist application and occurred without a detectable change in surface receptor expression (*Figure 1E*). Accordingly, both mGsi and Nb33 can be used as biosensors of ligand-dependent recruitment by KOR in living cells using the TIR-FM assay, and both sensors produce a reversible recruitment signal that is sufficiently robust and fast ($t_{1/2} < 30$ s) to enable reliable detection of protein recruitment without possible complications of later receptor trafficking.

We next tested two non-peptide KOR full agonists, U69593 (U69) and U50488 (U50). We generated concentration-response curves by increasing agonist concentration in a stepwise manner and then adding DynA in excess (10 µM) at the end of each series as an internal reference (*Figure 1G and H*). Both Nb33 and mGsi were robustly recruited in a concentration-dependent manner in response to DynA and both of the non-peptide full agonist drugs (*Figure 1I–K*), consistent with the previously established pharmacology of these compounds (*DiMattio et al., 2015*), but we also noted that the concentration-response relationship for mGsi recruitment was consistently left-shifted relative to Nb33. These results demonstrate that both Nb33 and mGsi are robustly recruited by KOR after activation by peptide and non-peptide full agonists in living cells, but with a potency shift indicating that the interactions are not identical.

## Agonist-selective recruitment of engineered protein probes

We then applied the same approach to investigate the effect of the alkaloid agonist etorphine (ET) on mGsi and Nb33 recruitment by KOR. ET is an opiate alkaloid drug that is structurally distinct from opioid peptides as well as from U50 and U69. ET efficaciously promotes G protein activation and signaling but has long been recognized to drive KOR internalization and phosphorylation poorly, supporting its classification as a G protein-biased agonist by operational criteria (*Chu et al., 1997*; *DiMattio et al., 2015*; *Jordan et al., 2000*). ET behaved as a potent but partial agonist in the mGsi recruitment assay, producing a maximum biosensor recruitment response reaching 67% of that produced by DynA (*Figure 2A and D*). Remarkably, ET produced little or no recruitment of Nb33 despite a robust response to DynA verified in each assay and in the same cells (*Figure 2B and E*). This lack of Nb33 recruitment was evident even at very high concentrations of ET (*Figure 2B and C*), in contrast to mGsi that was potently recruited (*Figure 2C–E*). Further verifying this difference, selective recruitment of mGsi relative to Nb33 was observed when the biosensors were tagged with distinct fluorophores, co-expressed, and imaged in parallel in the same cells (*Figure 2F*). Again, mGsi was potently recruited in response to ET but Nb33 was not, despite DynA producing strong

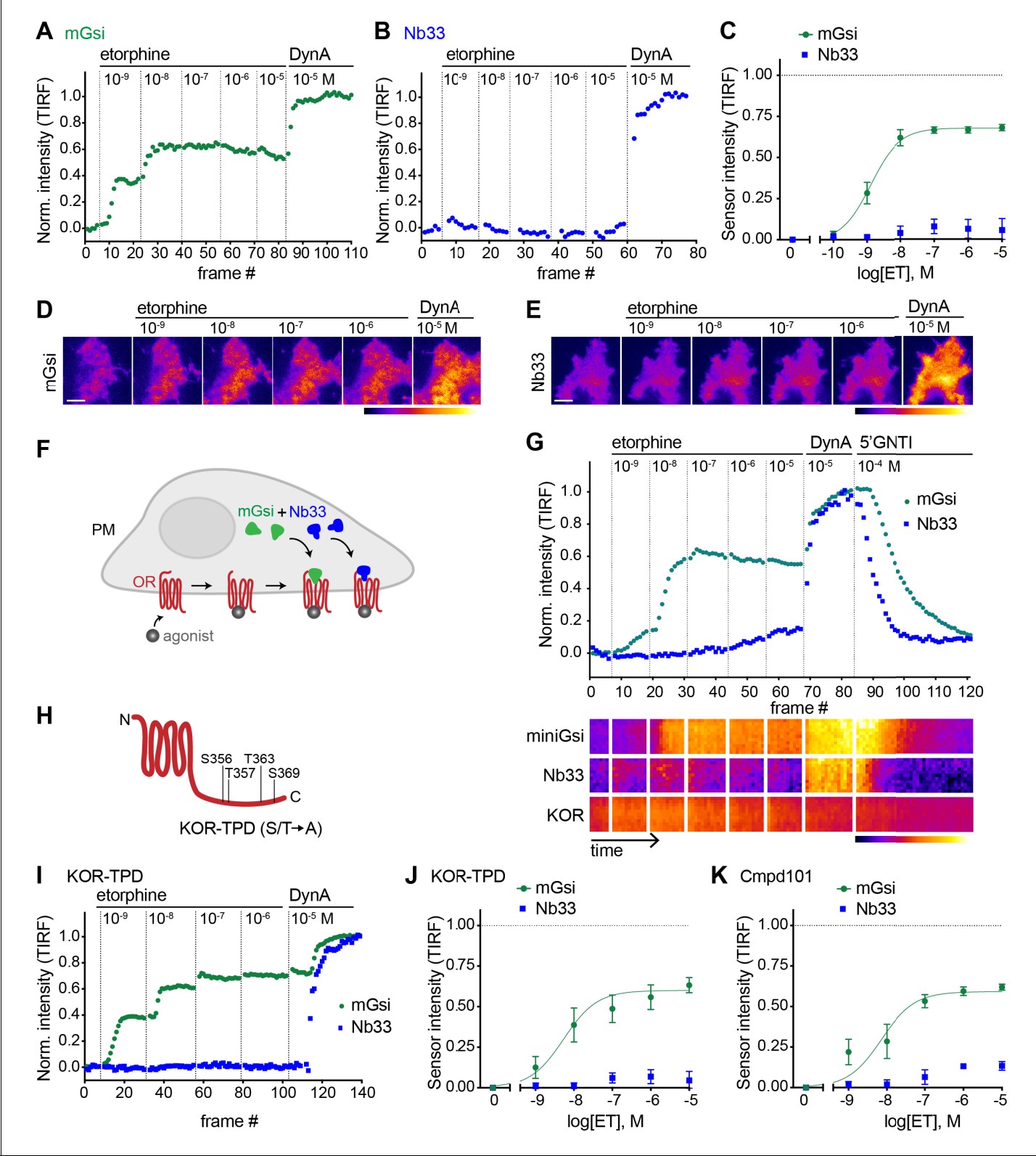

**Figure 2.** Selective recruitment of protein probes by KOR upon activation by etorphine. (**A**) mGsi intensity during TIR-FM time-lapse series of a HEK293 cell, co-expressing Venus-mGsi and KOR, adding increasing concentrations of etorphine (1 nM - 10 µM), followed by reference compound DynA (10 µM). 5 s between frames is shown. Intensity is normalized between 0 (no agonist) and 1 (reference compound DynA). (**B**) Nb33 intensity during TIR-FM time-lapse series of a HEK293 cell, co-expressing EGFP-Nb33 and KOR, treated, imaged, and normalized as in (**A**). (**C**) Concentration-dependent

*Figure 2 continued on next page*

*Figure 2 continued*

recruitment of mGsi and Nb33 probes to KOR upon etorphine (ET) addition, measured by TIR-FM and using DynA as reference. Normalization of intensity values is shown (range [0–1]). Regression curves with Hill slope of 1 are shown. n = 5; average ± SEM. (**D**) TIR-FM images of a time series of a HEK293 cell, expressing Venus-mGsi and KOR (not shown). Increasing concentrations of etorphine were added, followed by DynA. Venus-mGsi is pseudocolored, low to high intensity. The scale bar represents 10 μm. (**E**) Same as in (**D**) but with HEK293 cell expressing EGFP-Nb33 instead of mGsi. EGFP-Nb33 is pseudocolored, low to high intensity. The scale bar represents 10 μm. (**F**) Experimental set up for measuring agonist-dependent recruitment of both mGsi and Nb33 to KOR in same cell. (**G**) mGsi and Nb33 intensity during TIR-FM time-lapse series of a HEK293 cell, co-expressing Venus-mGsi, mCherry-Nb33, and FLAG-KOR. Cell was treated with increasing concentrations of etorphine, followed by DynA, and antagonist 5'GNTI. 5 s between frames is shown. Intensity is normalized between 0 (no agonist) and 1 (reference DynA). Lower panel: 10 min kymograph traced inside the cell, depicting intensities of Venus-mGsi, mCherry-Nb33, and FLAG-KOR (labeled with anti-FLAG M1-AF647), all pseudocolored, low to high intensity. (**H**) Schematic of the C-tail domain of KOR, indicating the known agonist-dependent phosphorylation sites that are mutated to alanine in KOR-TPD. (**I**) Same as in (**G**) but with HEK293 cell, co-expressing Venus-mGsi, mCherry-Nb33, and FLAG-KOR-TPD. (**J**) Concentration-dependent recruitment of mGsi and Nb33 probes to KOR-TPD upon etorphine addition. Experimental setup and analysis as in (**C**). n = 5; average ± SEM. (**K**) Concentration-dependent recruitment of mGsi and Nb33 probes to KOR upon etorphine addition, in cells pre-treated with GRK2/3 inhibitor Cmpd101 (30 μM). Experimental setup and analysis as in (**C**). n = 3; average ± SEM.

The online version of this article includes the following source data for figure 2:

**Source data 1.** ET concentration-dependent recruitment of mGsi and Nb33 probes to KOR, or to KOR-TPD, or to KOR in the presence of Cmpd101.

recruitment of both probes and in the same cells (*Figure 2G*). These results indicate that mG and Nb probes can distinguish receptor-proximal agonist effects in intact cells.

A simple interpretation of these results is that differential probe recruitment reflects a primary allosteric effect at the level of receptor-proximal protein engagement by the agonist-activated opioid receptor. An alternative possibility is that agonists produce differential probe recruitment as a secondary consequence of agonist-selective post-translational modifications of the receptor. In particular, because agonist-induced internalization of KOR requires multi-site phosphorylation on its cytoplasmic tail, and ET is known to stimulate this phosphorylation less strongly than DynA (*Chen et al., 2016*), we considered the possibility that differential biosensor recruitment occurs secondarily to differential phosphorylation. To test this, we measured biosensor recruitment by a mutant KOR lacking all relevant phosphorylation sites in the cytoplasmic tail (KOR-TPD for 'total phosphorylation defective', *Figure 2H*). The pronounced difference in mGsi relative to Nb33 recruitment was still observed (*Figure 2I and J*). Independently verifying this, selective probe recruitment by wild type KOR was not detectably perturbed in the presence of Compound101 (*Figure 2K*), a chemical inhibitor of GRK2/3 activity known to strongly reduce KOR phosphorylation in HEK293 cells (*Chiu et al., 2017*). Together, these results support the hypothesis that selective recruitment of mG relative to Nb probes occurs as a primary consequence of allosteric protein selection at the receptor, rather than a secondary effect of differential phosphorylation.

## Agonist-selective probe recruitment is not restricted to KOR

We next asked if our experimental strategy can also detect differential protein recruitment by MOR. Nb33 is already known to be recruited by agonist-activated MORs (*Stoeber et al., 2018*), and we verified that this is also the case for mGsi. DAMGO, a peptide full agonist of MOR, produced rapid and robust recruitment of mGsi that was rapidly reversed by the competitive antagonist naloxone (*Figure 3A and B*). Similar to what was observed for recruitment of the engineered protein probes by KOR, the concentration-response relationship for recruitment of mGsi by DAMGO was left-shifted relative to Nb33 (*Figure 3C*). ET (also an agonist of MOR) promoted recruitment of both probes by MOR, and to the same maximum degree when compared to the peptide full agonist (*Figure 3C*). This contrasts with partial recruitment of mGsi and no detectable recruitment of Nb33 by KOR (*Figure 2*), indicating that differential recruitment of the engineered protein probes by opioid receptors is both agonist-dependent and receptor subtype-specific.

To expand our search, and taking into account the fact that DAMGO and ET are both generally classified as full agonists at MOR, we next examined morphine and PZM21. Both of these non-peptide drugs are partial agonists with respect to assays of G protein activation or signaling, but each is derived from a different chemical scaffold and differs in degree of bias estimated using a beta-arrestin recruitment assay (*Manglik et al., 2016*). Using the same experimental protocol, and comparing recruitment promoted by the test ligand relative to the peptide full agonist (DAMGO)

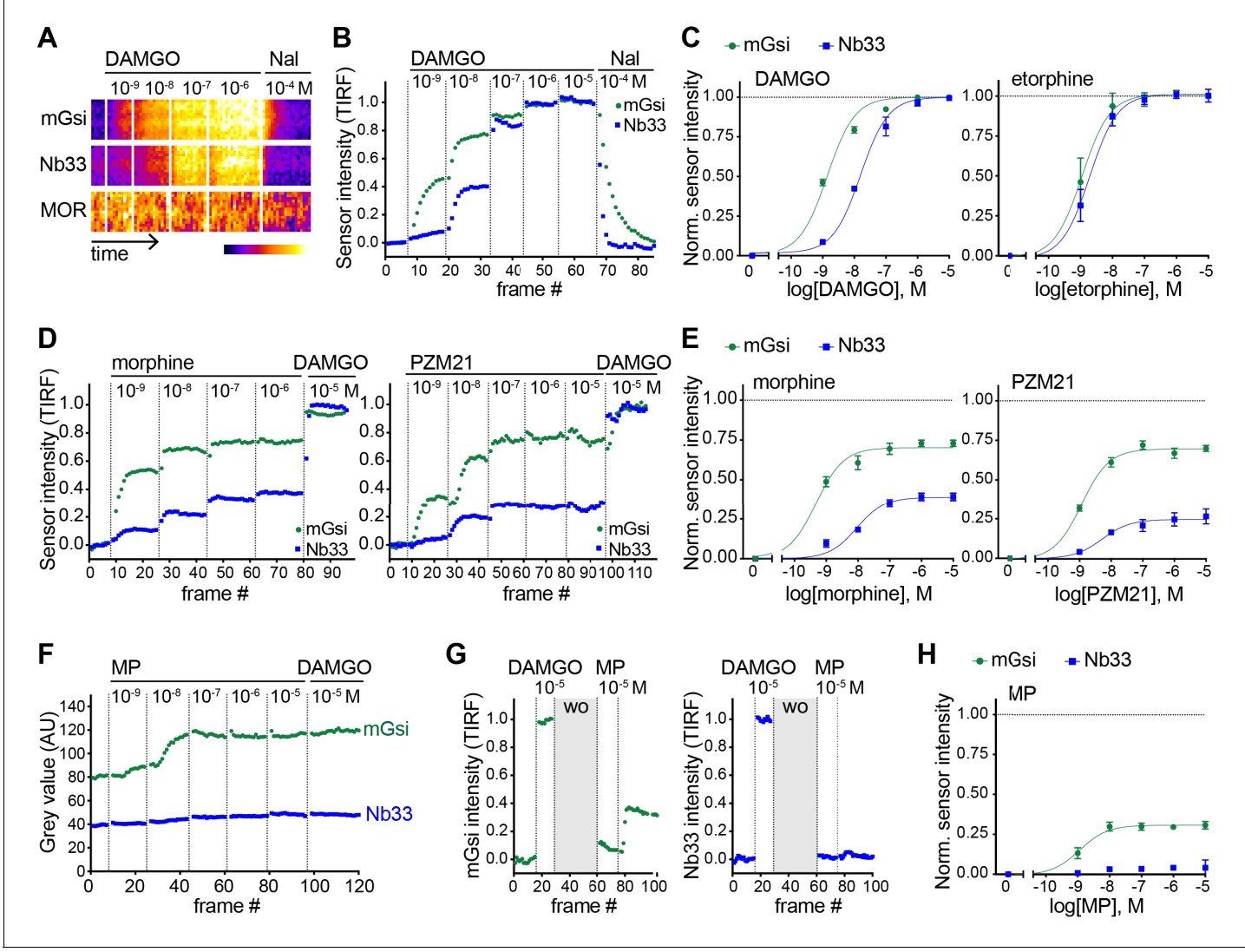

**Figure 3.** Agonist-selective protein probe recruitment by MOR. (**A**) 7 min kymograph traced inside a cell expressing Venus-mGsi, mCherry-Nb33, and FLAG-MOR (labeled with anti-FLAG M1-AF647) and treated with increasing concentrations of DAMGO (agonist), followed by addition of Naloxone (antagonist). Fluorescence intensities are pseudocolored, low to high intensity. (**B**) mGsi and Nb33 intensity during TIR-FM time-lapse series of a HEK293 cell, co-expressing Venus-mGsi, mCherry-Nb33, and FLAG-MOR, adding increasing concentrations of DAMGO followed by Naloxone. 5 s between frames is shown. Intensity is normalized between 0 (no agonist) and 1 (10 μM DAMGO). (**C**) Concentration-dependent recruitment of mGsi and Nb33 to MOR upon DAMGO and etorphine addition, measured by TIR-FM. Normalization of intensity values is shown (range [0–1]) with DAMGO as reference. Regression curves with Hill slope of 1 are shown. DAMGO n = 3, etorphine n = 4, average ± SEM. (**D**) mGsi and Nb33 intensity during TIR-FM time-lapse series of a HEK293 cell, co-expressing Venus-mGsi, mCherry-Nb33, and FLAG-MOR, adding increasing concentrations of morphine or PZM21 followed by reference compound DAMGO (10 μM). 5 s between frames is shown. Intensity is normalized between 0 (no agonist) and 1 (10 μM DAMGO). (**E**) Concentration-dependent recruitment of mGsi and Nb33 probes to MOR upon morphine or PZM21 treatment, setup and analysis as in (**C**). morphine n = 5, PZM21 n = 5, average ± SEM. (**F**) mGsi and Nb33 intensity during TIR-FM time-lapse series of a HEK293 cell, co-expressing Venus-mGsi, mCherry-Nb33, and FLAG-MOR, adding increasing concentrations of mitragynine pseudoindoxyl (MP) followed by DAMGO, using bath application. 5 s between frames is shown. (**G**) mGsi (left) and Nb33 (right) intensity during TIR-FM time-lapse series of a cell, co-expressing Venus-mGsi, mCherry-Nb33, and FLAG-MOR, adding 10 μM of reference DAMGO, followed by agonist washout using perfusion ('wo', highlighted in gray), and addition of 10 μM MP. 5 s between frames is shown. Intensity is normalized between 0 (no agonist) and 1 (10 μM DAMGO). (**H**) Concentration-dependent recruitment of mGsi and Nb33 to MOR upon MP addition, measured by TIR-FM with DAMGO as reference. n = 4; average ± SEM.

The online version of this article includes the following source data for figure 3:

**Source data 1.** Concentration-dependent recruitment of mGsi and Nb33 probes to MOR in response to DAMGO, ET, morphine, or PZM21.

reference, both morphine and PZM21 produced partial recruitment of mGsi as well as Nb33 (*Figure 3D and E*). Whereas morphine and PZM21 were similar in the degree of mGsi recruitment that they produced at saturating concentration, morphine was found to be significantly more efficacious than PZM21 in recruiting Nb33. Together, these results reveal a range of selective protein recruitment effects among chemically diverse MOR partial agonists.

The experimental strategy used to compare test agonist effects relative to the peptide reference was robust in practice but, in principle, it could underestimate differences relative to the reference peptide if the test agonist dissociates slowly or has an on-rate much faster than the peptide reference. We found evidence for this when evaluating another chemically distinct MOR partial agonist, the semi-synthetic natural product mitragynine pseudoindoxyl (MP) (*Váradi et al., 2016*). Using the sequential agonist addition protocol, MP appeared to be similarly efficacious to DAMGO in promoting recruitment of mGsi because no further increase was elicited by subsequent addition of DAMGO while, in contrast, MP failed to produce any detectable recruitment of Nb33. However, we noted that DAMGO also failed to promote recruitment of Nb33 in cells that were previously exposed to MP (*Figure 3F*), despite DAMGO promoting a strong Nb33 recruitment response in cells not previously exposed to MP (*Figure 3B*). Adding a perfusion wash step, in order to remove excess test agonist between applications, avoided this complication. With this modification, MP was verified to indeed promote mGsi recruitment by MOR, but to a significantly reduced maximal degree relative to DAMGO and without promoting detectable recruitment of Nb33 (*Figure 3G and H*). These results further expand the range of differential protein recruitment effects documented among chemically diverse MOR agonists.

## Differential protein recruitment can be elicited by diverse opioid agonists

To simplify comparison across agonists and receptors, we defined the maximum recruitment response elicited by each agonist compared to the corresponding peptide full agonist reference (DynA for KOR and DAMGO for MOR) as a relative 'intrinsic activity' for each agonist (*Figure 4A*). We then plotted these relative values for each biosensor (*Figure 4B*). Some non-peptide agonists were indistinguishable from the reference peptide by this analysis, recruiting both protein probes to

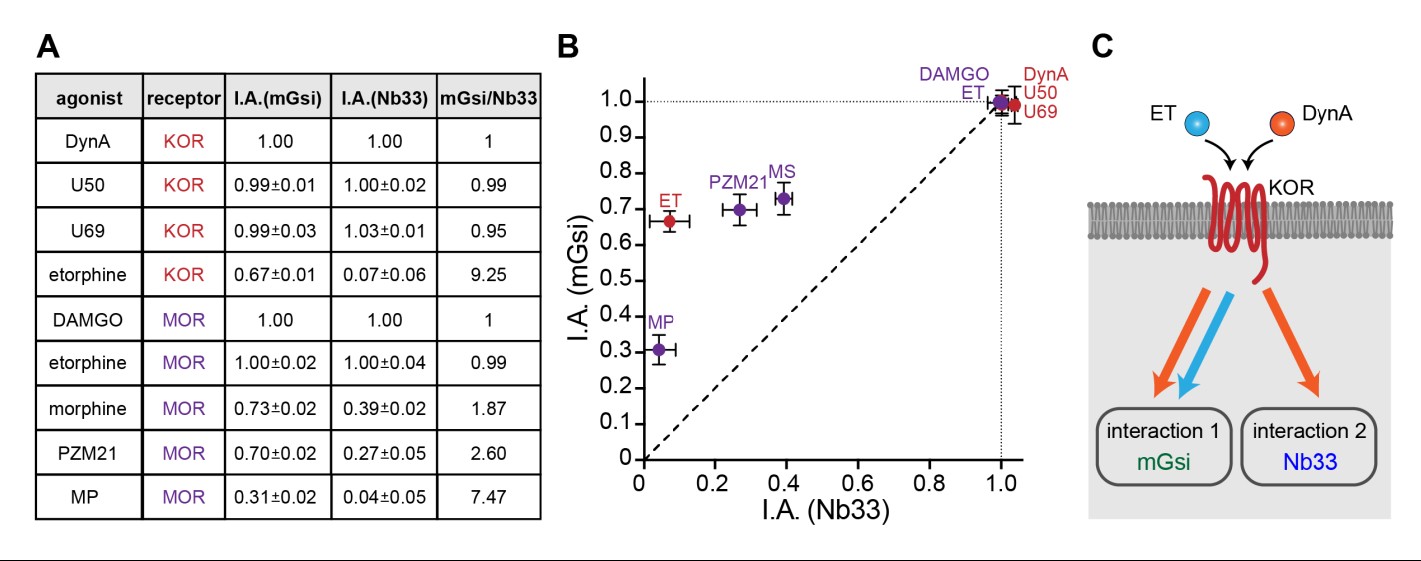

**Figure 4.** Receptor-proximal probe recruitment across agonists and receptors. (**A**) Table summarizing mGsi and Nb33 recruitment efficacies to KOR and MOR upon different agonists. Intrinsic activities ('I.A.', maximal response) for both probes and each agonist are given (average ± SEM). DynA serves as reference for KOR, DAMGO as reference for MOR. mGsi/Nb33 = ratios of intrinsic activities. (**B**) Plot of intrinsic activities (maximal responses) of mGsi recruitment as function of Nb33 recruitment for all KOR and MOR agonists. The diagonal (dotted) line indicates the theoretical trajectory for probe recruitment without bias. MS = morphine, ET = etorphine, MP = mitragynine pseudoindoxyl. (**C**) Summary of the effects of DynA vs. etorphine on KOR-proximal protein recruitment. The chemically distinct agonists differentially promote recruitment of protein probes mGsi (interaction 1) and Nb33 (interaction 2), revealing biased recruitment of cytoplasmic proteins by opioid receptors.

a similar maximal degree (corresponding to an 'I.A.' value of 1 for both probes), but others departed from the diagonal. This is not consistent with the traditional concept of partial agonism based on a unitary agonist-induced receptor 'on' state, which would predict the recruitment responses elicited by all agonists to fall along the diagonal. Rather, the present results support the view that opioid receptors are more flexibly activated, enabling them to selectively recruit one interaction probe over another in living cells. They further suggest that the ability to promote selective protein recruitment is widespread among chemically diverse opioid agonists (*Figure 4C*).

## Relevance to agonist-selective recruitment of GRK2

While we found the engineered proteins useful as orthogonal probes to unambiguously assess receptor-proximal recruitment in living cells, their disconnection from endogenous cellular machineries and pathways means that they are not directly related to function. Accordingly, we next asked if agonist-selective protein recruitment applies to a physiologically relevant GPCR-interacting protein. We focused on GRK2 because this kinase is known to be important for generating agonist-selective patterns of multi-site phosphorylation in the MOR cytoplasmic tail, which convey biased effects downstream from the receptor by distinguishing engagement of beta-arrestins and regulating receptor entry into the endocytic network (*Just et al., 2013*; *Lau et al., 2011*). We were also intrigued by GRK2 because it is recruited by activated GPCRs through multiple interactions, including with the activated GPCR and with beta-gamma subunits that are exposed on the inner membrane leaflet following activation of the G protein heterotrimer (*DebBurman et al., 1995*; *Lodowski et al., 2003*; *Figure 5A*).

We began by examining a functional GFP-fusion of GRK2 using the same TIR-FM imaging assay used to monitor orthogonal probe recruitment. We focused on comparing the effects of ET relative to DynA on KOR because these agonist-receptor pairs appeared to differ most dramatically based on the orthogonal biosensor recruitment assay (*Figure 4*). DynA promoted rapid, concentration-dependent recruitment of GRK2 to the plasma membrane (*Figure 5B*) while ET, despite being highly potent, produced a degree of GRK2 recruitment clearly lower than that produced by DynA (*Figure 5C and D*). This difference was not a secondary effect of receptor phosphorylation because ET also produced less maximal GRK2 recruitment than DynA using the phosphorylation-defective mutant KOR-TPD in place of KOR (*Figure 5E and F*).

Although ET promoted recruitment of full-length GRK2 less strongly than DynA, these agonists produced similarly strong recruitment of a probe corresponding to the isolated C-terminal PH domain from GRK2 that interacts with G beta-gamma (*Figure 5G*). This suggested that GRK2 binding to G beta-gamma subunits, enabled by G protein activation triggered by either agonist, is responsible for partial recruitment promoted by ET. We independently verified this conclusion by returning to assay of full length tagged GRK2, and testing the effect of blocking Gi activation by pre-exposing cells to pertussis toxin (PTX). In this condition, ET failed to produce any detectable recruitment of GRK2. However, as expected, DynA still produced a significant recruitment response in the same cells (*Figure 5H*), but to a reduced degree relative to the recruitment response elicited by DynA in cells not previously exposed to pertussis toxin.

The above results indicate that ET and DynA share the ability to promote GRK2 recruitment to the plasma membrane via binding G beta-gamma, and that DynA engages an additional mode of binding that is separate from the G protein and not shared with ET. We hypothesized that this interaction occurs with the activated opioid receptor itself. In order to test this, we devised an assay to resolve GRK2 recruitment to the plasma membrane from GRK2 binding directly to the receptor. To do so, we clustered receptors on the cell surface using an antibody cross-linking protocol, forming clusters that appeared in TIRF images as discrete spots of laterally concentrated KOR (*Figure 6A and B*, 'KOR' panels). We then used this characteristic appearance to distinguish GRK2 recruitment to KOR-containing clusters from recruitment to the surrounding plasma membrane separately from KOR clusters. As expected, in the absence of agonist GRK2 was primarily distributed in the cytosol and not detectably associated with KOR (*Figure 6B* left, 'GRK2' panel). Within ~1 min after application of DynA, GRK2 specifically accumulated at the KOR-containing clusters (*Figure 6B* right). In contrast, application of ET produced a diffuse increase in GRK2 fluorescence at the plasma membrane but no specific accumulation at KOR-containing clusters (*Figure 6C*). Quantification of the GRK2 intensity in KOR clusters relative to the surrounding plasma membrane verified significant accumulation of GRK2 with receptors promoted by DynA but not ET (*Figure 6D*), despite both agonists

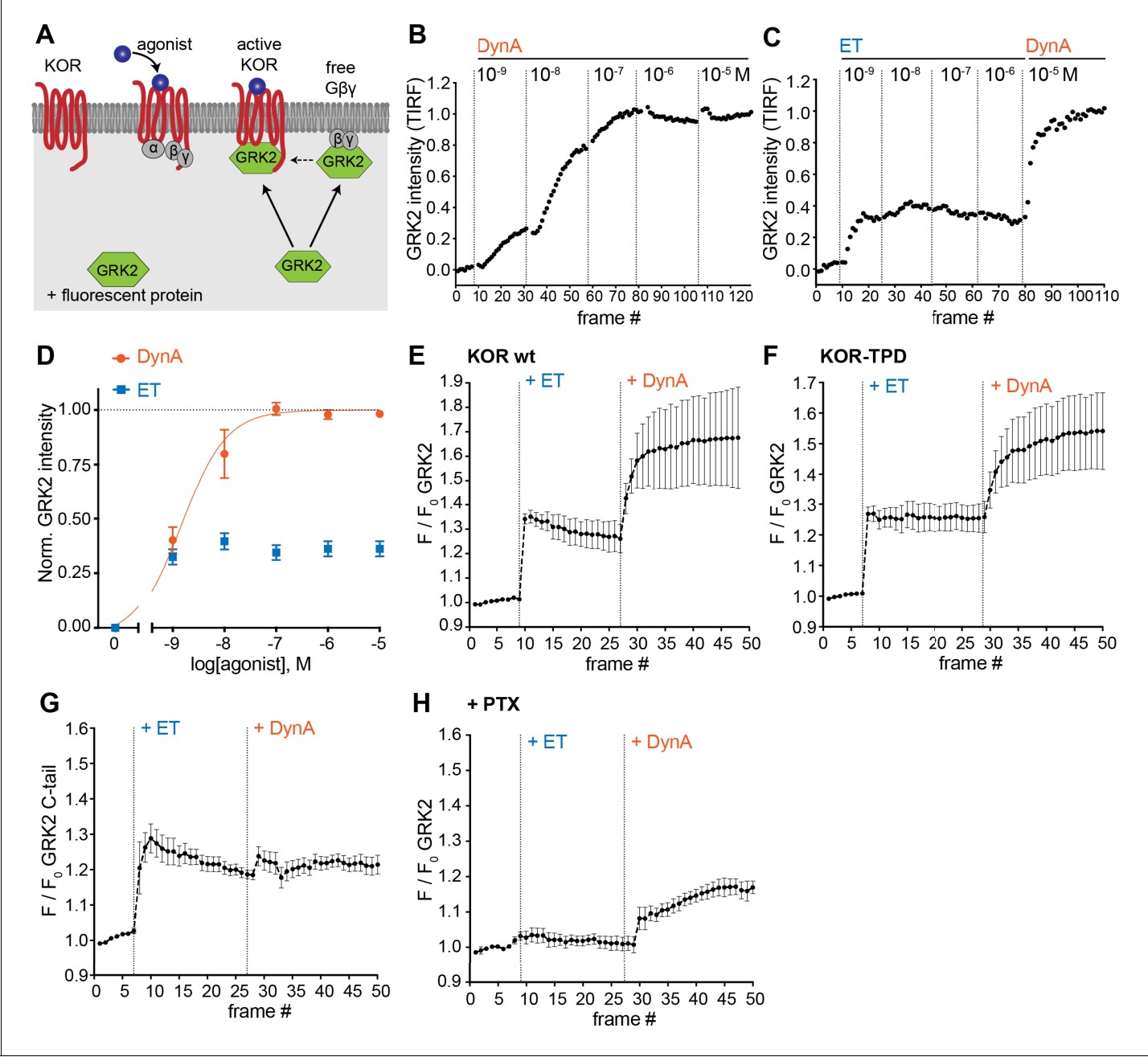

**Figure 5.** Agonist-selective modes of GRK2 recruitment to the plasma membrane. (**A**) Schematic depicting two modes of GRK2 recruitment from the cytosol to the plasma membrane upon KOR activation: one involves interaction with G beta-gamma subunits exposed upon G protein activation and another with the activated receptor itself. (**B**) GRK2 intensity during TIR-FM time-lapse series of a HEK293 cell, co-expressing GRK2-EGFP and KOR, adding increasing concentrations of DynA. 5 s between frames is shown. Intensity is normalized between 0 (no agonist) and 1 (10 µM DynA). (**C**) GRK2 intensity for cells transfected as in (**B**) but adding increasing concentrations of etorphine (ET), followed by 10 µM DynA as reference. Imaging and normalization as in (**B**). (**D**) Concentration-dependent recruitment of GRK2 to the plasma membrane upon DynA or ET addition, measured by TIR-FM. Normalization of intensity values is shown (range [0–1]) with DynA as reference. Regression curve with Hill slope of 1 is shown for DynA, no fit for ET. DynA n = 6, ET n = 7, average ± SEM. (**E**) Intensity of GRK2 during the TIR-FM time-lapse series of a HEK293 cell, expressing GRK2-EGFP and FLAG-KOR (not shown). Medium was exchanged to ET (100 nM) and then to DynA (1 µM) by bath application. 5 s between frames is shown. $F_0$ is the average fluorescence intensity before agonist. n = 4, average ± SEM. (**F**) GRK2 intensity time course as in (**E**), but cells express FLAG-KOR-TPD instead of wild-type. n = 4, average ± SEM. (**G**) Intensity of GRK2-C-tail during the TIR-FM time-lapse series of a HEK293 cell, expressing GRK2-C-tail-EGFP and FLAG-KOR, imaged and treated as in (**E**). n = 5, average ± SEM. (**H**) GRK2 intensity time course as in (**E**), but cells were pre-treated with pertussis toxin (PTX, 100 ng/ml). n = 3, average ± SEM.

*Figure 5 continued on next page*

*Figure 5 continued*

The online version of this article includes the following source data for figure 5:

**Source data 1.** Concentration-dependent recruitment of GRK2 to the plasma membrane in response to DynA or ET (*Figure 5D*).

**Source data 2.** Recruitment behavior of GRK2 to the plasma membrane in response to ET and DynA (*Figure 5E–H*).

promoting diffuse membrane recruitment (*Figure 5*). These results support a model of GRK2 engagement driven by discrete biochemical modes which are differentially regulated by agonists: DynA and ET share the ability to promote GRK2 recruitment to the plasma membrane via receptor-activated G beta-gamma, but DynA is different from ET in its ability to additionally promote GRK2 recruitment by binding directly to KOR (*Figure 6E*).

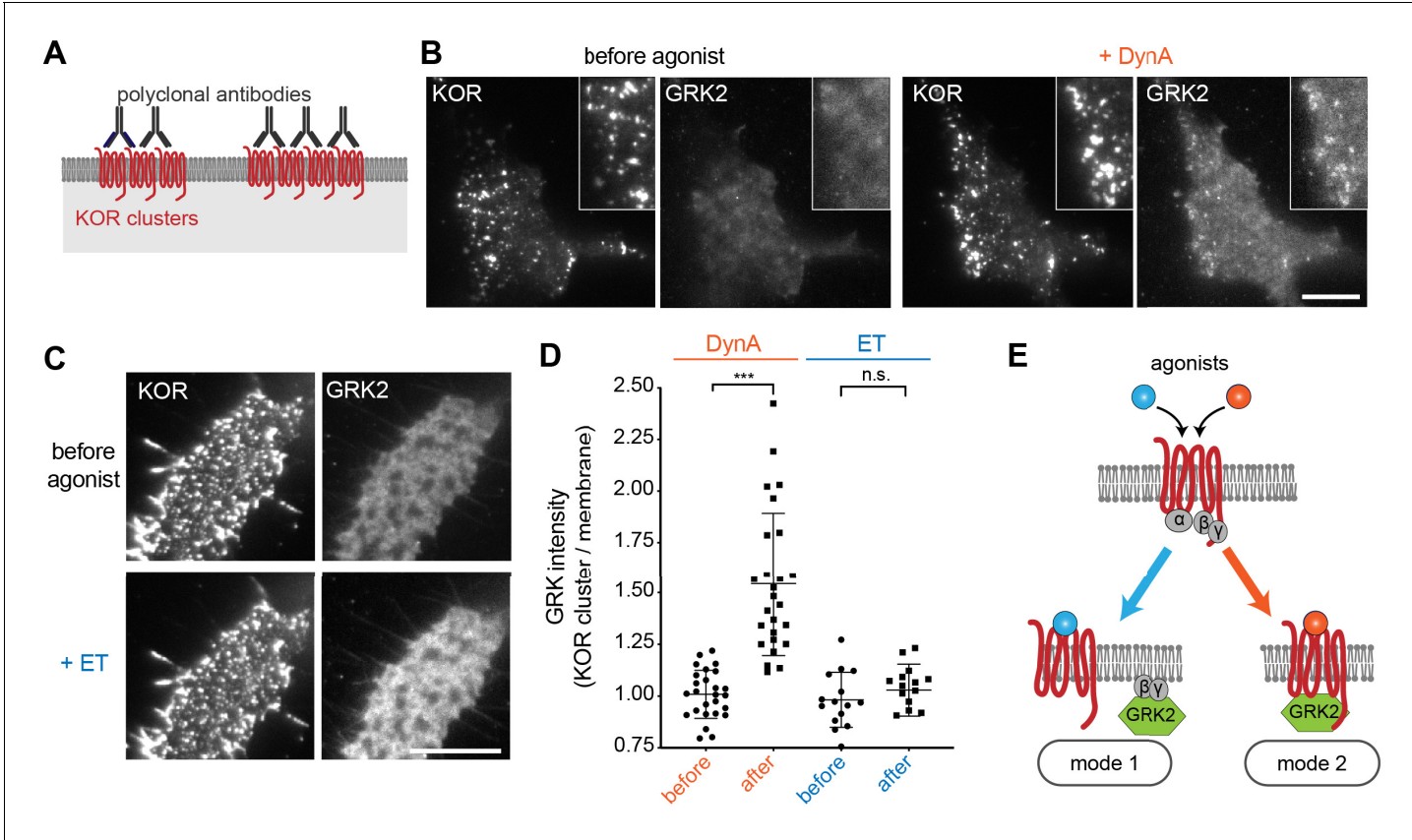

**Figure 6.** Agonist-selective modes of GRK2 recruitment by KOR. (**A**) Schematic of receptor clustering in the plasma membrane using cross-linking by polyclonal antibody. (**B**) TIR-FM images of a cell expressing GRK2-mCherry and SEP-KOR. KOR was cross-linked with polyclonal antibodies before imaging. Frames were collected immediately before agonist (left panels) and 1 min after application of 1 μM DynA (right panels). The scale bar represents 10 μm. (**C**) Same as in (**B**) but cells were treated with etorphine (ET, 1 μM) instead of DynA. The scale bar represents 10 μm. (**D**) GRK2 intensity in KOR clusters relative to surrounding plasma membrane (see methods). Quantification of images collected before agonist and 2 min after agonist (DynA or ET) application, similar to images shown in (**A**) and (**B**). DynA (26 cells) and ET (15 cells) across three independent experiments. Mean with SD is shown. ***p = <0.0001 by paired two-tailed t-test. n.s. = not significant (p=0.30). (**E**) The two biochemical modes of GRK2 recruitment are selectively promoted by distinct agonists. While etorphine only drives GRK2 binding to G beta-gamma, DynA additionally promotes direct interaction of GRK2 with activated KOR.

The online version of this article includes the following source data for figure 6:

**Source data 1.** Recruitment of GRK2 to KOR clusters upon DynA or ET treatment (*Figure 6D*).

# Discussion

The ability of agonists to impose selectivity on protein recruitment by GPCRs has been proposed for many years and is a core hypothesis underlying the present concept of biased agonism (*Schmid et al., 2017*; *Smith et al., 2018*; *Urban et al., 2007*), but testing this hypothesis in an intact cellular environment has remained challenging due to the complexity of cellular transduction and regulatory pathways that GPCRs typically engage (*Kenakin, 2019*). The present study describes a direct, reductionist approach to this problem based on the application of engineered proteins that bind activated receptors but are not known to bind other cellular proteins. We show that agonists differ in relative ability to drive recruitment of the engineered probes to opioid receptors in living cells, and then delineate how the principle of agonist-selective recruitment applies to GRK2 as a physiologically relevant regulator.

Our results indicate that selective recruitment of one cellular protein over another not only occurs in intact cells, but it is widespread and elicited by diverse agonists. All partial agonists examined were found to promote mGsi recruitment more strongly than Nb33 when present at saturating concentration. Further, concentration-response curves for mGsi relative to Nb33 recruitment by opioid receptors were left-shifted even for peptide full agonists. The allosteric nature of GPCR activation is well established, and has been recognized since early studies of receptor coupling to heterotrimeric G proteins in vitro (*De Lean et al., 1980*; *Maguire et al., 1975*; *Strachan et al., 2014*; *Sunahara and Insel, 2016*). The present results are fully consistent with this concept, and expand it by providing clear biochemical evidence for discrete protein-engaged receptor states that can be selectively produced by diverse agonists in the complex environment of intact, living cells. The engineered interaction probes that we focused on here demonstrate such an additional level of allosteric selection most simply, but our results delineating differential recruitment of GRK2 by receptors suggest that the same principle applies in a more complex manner to biologically relevant GPCR-interacting proteins.

In its present state of development, our approach is limited by the number of orthogonal probes available for assessing protein recruitment. We focused here on two previously validated GPCR-interacting proteins, selected based on existing biophysical evidence that each recognizes different structural features of the activated receptor. It is possible, and we think likely, that still more specificity exists in receptor-proximal protein recruitment. In future studies it will be interesting to develop or adapt additional structurally diverse protein folds to address this question, and to explore additional agonist diversity using the existing probes. For example, it will be interesting to determine if ligands can be found that promote recruitment of Nb33 preferentially to mGsi.

An important next step is to delineate the biophysical basis for the observed selectivity of protein recruitment induced by opioid agonists, with differential effects of DynA and ET on Nb33 recruitment to KOR providing a striking example. The present results clearly indicate that the complexes responsible for agonist-selective protein recruitment must be distinct, but leave unresolved the nature of the distinction. One possibility is that distinct allosteric complexes reflect unique conformational ensembles of the receptor. Although currently available structural data for KOR (*Che et al., 2018*; *Wu et al., 2012*) preclude direct assessment of such conformational differences, prior structures of MOR in complex with either Nb39 (*Huang et al., 2015*) (a close analog of Nb33) or heterotrimeric Gi (*Koehl et al., 2018*) offer some insight. Both Nb39 and Gi alpha stabilize an active MOR conformation in the intracellular domain; however, the precise conformation of the MOR intracellular loop 3 (ICL3) differs between the two structures. Thus, differential recruitment of Nb33 and mGsi may reflect agonist-selective stabilization of distinct active receptor conformations. Alternatively, agonists may promote receptors to adopt similar active conformations, and distinctions in the kinetics of sensor binding, sensor concentration, and/or sensor affinity contribute to differential recruitment. Future studies, combining biophysical and cell biological approaches, will be needed to answer this question. We also note that these classes of mechanism are not mutually exclusive, and think it is likely that both contribute to agonist-selective allosteric effects observed in intact cells.

It will be particularly interesting to extend the present approach toward examining kinetic aspects of selective protein recruitment by receptors. We found that the orthogonal probes produce a time-invariant recruitment response within ~30 s after agonist application. This enables the approach to be used as an end-point assay scalable to a drug screening platform, and we focused on steady state recruitment values in the present work for simplicity. However, in light of clear and long-standing

evidence for kinetic differences in agonist action at GPCRs (*Klein Herenbrink et al., 2016*; *Swaminath et al., 2004*), we anticipate that time-dependent analysis of probe recruitment will provide additional insight into selectivity among agonists.

In sum, and viewed more broadly, the present results reinforce an emerging understanding that GPCRs operate as allosteric machines with the potential to communicate significantly more information about local chemical environment than the mere presence or absence of a cognate agonist (*Costa-Neto et al., 2016*; *Kenakin, 2019*). We propose from the present observations that the mGsi probe reports allosteric effects relevant to G protein engagement by opioid receptors, and that the Nb probe reports additional effects relevant to GRK engagement. Our results further support the hypothesis that agonist bias, now generally defined by operational criteria, can be deconvolved into discrete receptor-proximal molecular selection events. The present study makes initial inroads toward decoding this underlying 'machine language' of GPCR signaling, and thus toward precisely delineating how much chemical information content receptors actually convey physiologically.

# Materials and methods

## Key resources table

| Reagent type (species) or resource | Designation | Source or reference | Identifiers | Additional information |
|---|---|---|---|---|
| Cell line (human, female) | HEK293 | ATCC | CRL-1573; RRID: CVCL_0045 | Human embryonic kidney |
| Antibody | Mouse anti-FLAG (M1) | Sigma-Aldrich | F-3040; RRID: AB_439712 | (1:1000) |
| Antibody | Rabbit anti-GFP | Invitrogen | A-11122; RRID: AB_221569 | (1:100) |
| Recombinant DNA reagent | EGFP-Nb33 | (*Stoeber et al., 2018*) | N/A | EGFP-C1 backbone |
| Recombinant DNA reagent | pmApple-Nb33 | (*Stoeber et al., 2018*) | N/A | pmApple-C1 backbone |
| Recombinant DNA reagent | NES-Venus-mGsi | (*Wan et al., 2018*) | N/A | pcDNA3 backbone |
| Recombinant DNA reagent | NES-Venus-mGs | (*Wan et al., 2018*) | N/A | pcDNA3 backbone |
| Recombinant DNA reagent | signal sequence FLAG (ssf)-MOR, murine | (*Stoeber et al., 2018*) | N/A | pcDNA3 backbone |
| Recombinant DNA reagent | ssf-KOR, murine | (*Chu et al., 1997*) | N/A | pcDNA3 backbone |
| Recombinant DNA reagent | ssf-KOR-TPD (S356A, T357A, T363A, S369A) | This study | N/A | pcDNA3 backbone, see Materials and methods |
| Recombinant DNA reagent | Super ecliptic pHluorin (SEP) -KOR, murine | This study | N/A | pCAGGS-SE backbone, see Materials and methods |
| Recombinant DNA reagent | GRK2-EGFP, murine | This study | N/A | pCAGGS-SE backbone, see Materials and methods |
| Recombinant DNA reagent | GRK2-pmApple, murine | This study | N/A | pCAGGS-SE backbone, see Materials and methods |
| Recombinant DNA reagent | GRK2-C-tail- EGFP (aa 546–670) | This study | N/A | pCAGGS-SE backbone, see Materials and methods |
| Peptide, recombinant protein | Dynorphin A (1–17, DynA) | Anaspec | AS-24298 | |
| Peptide, recombinant protein | DAMGO, [D-Ala2, N-Me-Phe4, Gly5-ol]-Enkephalin acetate salt | Sigma-Aldrich | E7384 | |
| Chemical compound, drug | U-69593 (U69) | Cayman Chemical | 13255 | |

*Continued on next page*

*Continued*

| Reagent type (species) or resource | Designation | Source or reference | Identifiers | Additional information |
|---|---|---|---|---|
| Chemical compound, drug | U-50488 hydrochloride (U50) | Tocris | 0495 | |
| Chemical compound, drug | GNTI dihydrochloride (5'GNTI) | Axon Med Chem | 1226 | |
| Chemical compound, drug | Etorphine-HCl | NIDA | N/A | |
| Chemical compound, drug | Morphine sulfate (MS) | Sigma-Aldrich | 1448005 | |
| Chemical compound, drug | Naloxone hydrochloride dihydrate | Sigma-Aldrich | N7758 | |
| Chemical compound, drug | PZM21 | Enamine | N/A | custom synthesis |
| Chemical compound, drug | mitragynine pseudoindoxyl (MP) | This study | N/A | (*Váradi et al., 2016*) |
| Chemical compound, drug | Compound101 (Cmpd101) | HelloBio | HB2840 | |
| Chemical compound, drug | Pertussis toxin | Sigma-Aldrich | P7208 | |
| Commercial assay or kit | Alexa Fluor 647 Protein Labeling Kit | Thermo Fisher Scientific | A20173 | |
| Software, algorithm | Prism | GraphPad | 8.1.1 | |
| Software, algorithm | ImageJ | Imagej.net/ contributors | 2.0.0-rc-54/1.51 g | |
| Software, algorithm | MATLAB | MathWorks | R2014b | |
| Software, algorithm | PyMOL | Schrödinger | v1.7.4.5 | |

## Mammalian cell culture conditions

HEK293 (CRL-1573, ATCC, female, mycoplasma-tested) were cultured in Dulbecco's modified Eagle's medium (DMEM, GIBCO), supplemented with 10% fetal bovine serum (UCSF Cell Culture Facility). Stably transfected HEK293 cells expressing N-terminally FLAG-tagged MOR or KOR were cultured in the presence of 250 µg/ml Geneticin (Gibco). For transient DNA expression, Lipofect-amine 2000 (Invitrogen) was used according to manufacturer's instructions. For live cell imaging, cells were plated on poly-L-lysine-coated 35 mm glass-bottomed culture dishes (MatTek Corpora-tion) 48 hr before the experiments. Cells were transfected 24 hr prior to imaging. Per 35 mm culture dish, 200 ng DNA was used for mGsi and Nb33, 300 ng DNA was used for GRK2 constructs and 1.2 µg DNA was used for receptor constructs.

## cDNA constructs

GRK2-EGFP and GRK2-pmApple were created by amplifying murine GRK2 and GFP or pmApple DNA by PCR and inserting GRK2 and the respective fluorescent protein using In-Fusion cloning into pCAGGS-SE cut with KpnI and EcoRI. Super ecliptic pHluorin (SEP)-KOR was generated by PCR amplification of SEP and KOR, and insertion using In-Fusion cloning into pCAGGS-SE cut with KpnI and EcoRI. ssfKOR-TPD was generated by In-Fusion cloning of three PCR fragments that cover ssfKOR and introduce mutations S356A, T357A, T363A, and S369A.

## Live cell total internal reflection fluorescence microscopy (TIR-FM)

Live cell image series measuring protein recruitment to the plasma membrane were performed at 37°C using a Nikon Ti-E microscope equipped for through-the-objective TIR-FM with a temperature-, humidity- and $CO_2$-controlled chamber (Okolab), objective heater, perfect focus system, and an Andor DU897 EMCCD camera. Images were obtained with a 100 × 1.49 NA Apo TIRF objective

(Nikon) with solid-state lasers of 488, 561 and 647 nm (Keysight Technologies). Before imaging, receptors at the cell surface were labelled with M1 monoclonal FLAG antibody (1:1,000) conjugated to Alexa647 dye for 10 min at 37°C. Cells were then washed and live imaged in HBS imaging solution (Hepes buffered saline (HBS) with 135 mM NaCl, 5 mM KCl, 0.4 mM MgCl2, 1.8 mM CaCl2, 20 mM Hepes, 5 mM d-glucose adjusted to pH 7.4 and 300–315 mOsmol/l). Agonists or antagonists were either added by bath application at concentrations indicated in the figure legends or by media perfusion. For the latter, an insert was 3D-printed and placed inside the imaging dish where it left a dead volume of about 300 µL. It was used to perfuse HBS imaging solution with agonists or without agonists (agonist washout) at concentrations indicated in the figure legends with a flow rate of 1.5 ml/min.

### Cell treatments prior to live cell imaging

To cluster receptors in the plasma membrane, cells transfected with SEP–KOR were treated with a polyclonal rabbit anti-GFP antibody (1:100) for 15 min at 37°C. Cells were then washed and imaged live in HBS imaging solution. To inhibit GRK2/3, cells were pre-incubated with Compound101 (30 µM) for 15 min at 37°C and Compound101 was present throughout the imaging experiment. To inhibit KOR coupling to Gαi/o, cells were treated with PTX (100 ng/mL) for 16 hr and PTX was present throughout the imaging experiment.

### Agonist concentration dependence of protein recruitment

For probing protein recruitment to the plasma membrane, HEK293 cells co-expressing the cytosolic protein of interest (mGsi, Nb33, or GRK2) and MOR or KOR were imaged using TIR-FM. Cells were treated with increasing concentrations of agonist (bath application) and imaged at a frame rate of 0.2/s (total movie length 6–8 min). Protein intensity during time lapse series was measured using ImageJ. If indicated, values were normalized between 0 (before agonist) and 1 (10 µM reference agonist). Regression curves with Hill slope of 1 were fit using Prism 8.

### Quantitative image analysis

All quantitative image analysis was performed on unprocessed images using MATLAB (MathWorks, R2014b) or ImageJ (2.0.0). For quantifying GRK2-mCherry recruitment to the plasma membrane and receptor clusters, we used a custom written MATLAB script. In brief, a polygon was drawn on the TIR-FM image to encompass the cell of interest. Then, a mask of the receptor clusters was generated by thresholding the SEP-KOR signal within the polygon. The average GRK2-mCherry fluorescence was measured within the cluster mask (KOR clusters) and outside of the mask (membrane) of the polygon, allowing to calculate the ratio. Quantification was performed in cells imaged before (t = 0) and after (t = 1–2 min) agonist addition.

### Statistics

Quantification of data are presented as mean ± standard error of the mean (SEM) or standard deviation of the mean (SD) based on at least three biologically independent experiments with the precise number indicated in the figure legends. Statistical analysis was performed using Prism (8.1.1, Graph-Pad) and using paired or unpaired two tailed Student's t test.

## Acknowledgements

Imaging experiments were carried out in the UCSF Nikon Imaging Center directed by DeLaine Larsen. We thank the group of Jean Braun at the German Research Center for Geosciences for a short-term stay of MS. The study was supported by research grants from the NIH (DA010711 and DA012864 to MvZ, OD023048 to AM, GM130142 to NAL, and R21DA045884 to SM). MS is supported by the Swiss National Science Foundation (P300PA_164712 and PCEFP3_181282). AM is supported by the Searle Scholars Program.

## Additional information

### Funding

| Funder | Grant reference number | Author |
|---|---|---|
| National Institutes of Health | DA010711 | Mark Von Zastrow |
| National Institutes of Health | DA012864 | Mark Von Zastrow |
| National Institutes of Health | OD023048 | Aashish Manglik |
| National Institutes of Health | GM130142 | Nevin A Lambert |
| Swiss National Science Foundation | P300PA_164712 | Miriam Stoeber |
| Swiss National Science Foundation | PCEFP3_181282 | Miriam Stoeber |
| Sears | | Aashish Manglik |
| National Institutes of Health | R21DA045884 | Susruta Majumdar |

The funders had no role in study design, data collection and interpretation, or the decision to submit the work for publication.

### Author contributions

Miriam Stoeber, Conceptualization, Data curation, Formal analysis, Methodology, Writing - original draft; Damien Jullié, Formal analysis, Methodology; Joy Li, Data curation, Formal analysis; Soumen Chakraborty, Susruta Majumdar, Nevin A Lambert, Resources; Aashish Manglik, Conceptualization, Resources; Mark von Zastrow, Conceptualization, Supervision, Funding acquisition, Methodology, Writing - original draft

### Author ORCIDs

Miriam Stoeber (ID) https://orcid.org/0000-0002-5210-2864
Nevin A Lambert (ID) http://orcid.org/0000-0001-7550-0921
Mark von Zastrow (ID) https://orcid.org/0000-0003-1375-6926

### Decision letter and Author response

Decision letter https://doi.org/10.7554/eLife.54208.sa1
Author response https://doi.org/10.7554/eLife.54208.sa2

## Additional files

### Supplementary files

• Transparent reporting form

### Data availability

All data generated or analysed during this study are included in the manuscript.

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
