## [Decision Letter]

**Acceptance summary:**

This manuscript examines agonist selective recruitment of engineered molecules directly to opioid receptors, both kappa and mu subtypes. The two engineered molecules, Nb33 and mGsi, have been previously well characterized and are used as surrogates for GRK and G-beta-gamma proteins. The results show that different opioid agonists differentially recruit the receptor association of the two sensor molecules independently of downstream signaling. These results are of considerable interest and constitute a major advance in our ability to understand GPCR signalling.

**Decision letter after peer review:**

Thank you for submitting your article "Agonist-selective recruitment of engineered protein probes and of GRK2 by opioid receptors in living cells" for consideration by *eLife*. Your article has been reviewed by two peer reviewers, including Volker Dötsch as the Reviewing Editor and Reviewer #1, and the evaluation has been overseen by Olga Boudker as the Senior Editor. The following individual involved in review of your submission has agreed to reveal their identity: John T Williams (Reviewer #2).

The reviewers have discussed the reviews with one another and the Reviewing Editor has drafted this decision to help you prepare a revised submission.

Summary:

The manuscript is well written and the work well executed, with many controls reported. Only one minor comment was raised during the discussion.

The only further and useful addition would be a consideration of the proposed structural changes induced by the different agonists in the Discussion section. Since crystal structures are known, some speculation about the differently activated states would add a mechanistic/structural component to the manuscript.

---

## [Author Response]

Summary:The manuscript is well written and the work well executed, with many controls reported. Only one minor comment was raised during the discussion.The only further and useful addition would be a consideration of the proposed structural changes induced by the different agonists in the Discussion section. Since crystal structures are known, some speculation about the differently activated states would add a mechanistic/structural component to the manuscript.

We are pleased that the reviewers appreciate the findings and consider them important for understanding GPCR signal transduction.

The reviewers raise an important point and we have included a new paragraph in the revised Discussion section that focuses on the possible structural / biophysical basis for the observed selectivity of protein recruitment to opioid receptors promoted by different agonists (please see the fourth paragraph of the Discussion section):

First, we discuss that distinct allosteric complexes may reflect unique receptor conformations. More specifically, available structures of MOR in complex with either nanobody or heterotrimeric Gi show differences in the conformation of intracellular loop 3 (ICL3). Therefore, differential recruitment of Nb and mGsi may reflect agonist-selective stabilization of distinct active receptor conformations, possibly involving distinct ICL3 arrangements.

Second, we outline the possibility that agonists may promote receptors to adopt similar conformations but that distinct sensor characteristics (kinetics of sensor binding, sensor concentration, and/or sensor affinity) contribute to the differential recruitment.

We think it is likely that both mechanisms contribute to the agonist-selective allosteric effects that we observe in intact cells and may similarly be used by endogenous GPCR interaction partners.